# What influences parental decisions about antibiotic use with their children: A qualitative study in rural Australia

**Stephanie A. Marsh, Mitchell K. Byrne●\*, Sara Parsafar**

Faculty of Health, Charles Darwin University, Darwin, Northern Territory, Australia

\* mitchell.byrne@cdu.edu.au

## Abstract

### Background

Inappropriate use of antibiotics can promote bacterial resistance, which is a growing public health concern. As children are amongst the highest recipients of antibiotics, understanding the drivers of parental decisions towards their children's antibiotic use is imperative for the development of strategies to assist parents in making more informed decisions. This is particularly relevant to the decisions of parents living in resource-limited settings. This study explored the perspectives and practices of parents living in a rural setting about the use of antibiotics in their children.

### Methods

Three focus group interviews were conducted with 10 parents of children under 12yrs, living in rural and remote areas of the Northern Territory, Australia. A focus group guide was developed to facilitate discussions, which was informed by the Theory of Planned Behaviour. Transcripts were analysed abductively using thematic analysis.

### Findings

Four themes emerged explaining factors that contributed to parent decisions about their children's antibiotic use: 1) Parental knowledge, attitudes and decision making; 2) Perceptions of antimicrobial resistance; 3) Healthcare challenges; 4) Behaviours with antibiotics. While parents demonstrated accurate knowledge of the indications for antibiotic use, their decisions about the need for antibiotics were often driven by fear of serious illness. This fear was exacerbated by resource limitations associated with living in a resource-limited setting. Additional drivers of parental decision making included what parents have read or heard from both medical and non-medical networks, underscoring the importance of Social Norms in predicting behavioural intentions.

### Conclusion

Mothers living in remote areas experiencing reduced access to health services may make decisions about antibiotic use out of fear and based on the advice of their personal network

**Data Availability Statement:** Data cannot be shared publicly because of of the potential to re-identify participants and thus breach confidentiality. Data are available from the Charles Darwing

University Institutional Ethics Committee (contact Professor Marilynne N Kirshbaum, +61 8 8946 6063) for researchers who meet the criteria for access to confidential data.

**Funding:** This study was part of a PhD study, funded by the Australian Government Research Training Program Scholarship. The funding source had no input into the design of the study, the collection, analysis or interpretation of the date or the writing of the manuscript.

**Competing interests:** NO authors have competing interests.

when they perceive their child is vulnerable to a health threat. Findings from this study provide guidance for future research in the prediction of antibiotic use behaviours and for context-specific interventions.

## Introduction

The growth in drug resistant bacteria is an increasing global public health issue. A key driver in the acceleration of antimicrobial resistance (AMR) is the overuse and misuse of antibiotics [1]. Children are particularly susceptible to infections and have a high rate of antibiotic use. Thus, the role of parents, who are the primary decision makers in the acquisition and administration of medication, is important in understanding the appropriate use of antibiotics in children [2].

International studies demonstrate that parents may persist in their request for antibiotic prescriptions from prescribers, not adhere to treatment instructions, or may use antibiotics without consulting a doctor [1–9]. These behaviours are considered sub-optimal and can facilitate the development of antibiotic resistant bacteria [10]. Parent behaviour with antibiotics has been shown to be influenced by their knowledge and beliefs [10], the perceived severity of the child's illness [11], the availability of antibiotics [12], the child's age [12–14] and the context [15]. From a contextual perspective, several studies have found a strong associated between poor access to healthcare and parent self-directed antibiotic use [16–20]. Access to health care is often challenged in rural and remote locations [21], a contextual variable likely to affect parental decisions about meeting the health care needs of their children [19]. Parents have also been found to make decisions about their children's antibiotic use based on advice from their social group or support network [1, 11, 18, 22].

However, there is an absence of theory informed research when examining parental perceptions about their children's antibiotic use [11, 12, 22–24]. Studies that have used qualitative methods have primarily been concentrated in Europe, the United States, Asia and Africa [22, 25–29]. There is currently a scarcity of qualitative research examining Australian parent views and behaviours regarding antibiotic use and antibiotic resistance [23, 30]. Mixed methods research indicates that parents living in an Australian capital city report they do not expect antibiotics to be prescribed for their children when visiting the General [Medical] Practitioner (GP) [31]. The research indicates that Australian consumers with consistent and reliable access to the same GP or medical practice, are more likely to possess shared expectations with their GP about minimising antibiotic use [32]. However, little is known about what guides decisions concerning antibiotic use amongst Australian parents with limited access to healthcare, especially those living in rural and remote areas.

This study explored the opinions and practices of parents living in remote areas of the Australian Northern Territory with respect to the use of antibiotics with their children. A critical variable of interest was the impact of limited access to health care on parental decision making, where limited access related to ease of obtaining medical appointments, consistency in provider, or distance to travel to such appointments. The research was both exploratory and theory informed, using the Theory of Planned Behaviour (TPB) as a guide in the understanding of behaviour. The TPB framework proposes that there are three core factors relevant to engaging in a given behaviour: *attitude* towards the behaviour; *subjective norms* in relation to the behaviour; and *perceived behavioural control* relating to the ease or difficulty of engagement in the behaviour [33]. These factors are thought to influence intention to engage in a particular

behaviour: in this case the appropriate or inappropriate use of antibiotics. The TPB has been used as a theoretical model in previous qualitative studies examining the use of antibiotics [34]. This appears to be the model's first application to research seeking to understand factors that influence parental decisions about their children's antibiotic use.

Previous international research cited above prompted the exploration of the following research questions with respect to rural and remote parents:

1. Does knowledge and beliefs about antibiotics influence parental attitudes regarding the use of antibiotics with their children?

2. Does limited access to primary health services lead to the use of social networks for medical advice regarding child antibiotic use?

3. Does limited access to primary health services contribute to parental decisions to pressure GPs for antibiotic prescriptions?

4. Does limited access to primary health services influence parental decisions to store/share antibiotics?

## Methods

### Study design

Focus group interviews were the chosen form of data collection. Focus groups enable the exploration of an issue that has limited background data or research and support building group dynamics to explore issues in context [35]. Our groups were mini focus groups. Mini groups consist of 2–5 participants and can be used when there is a small participant pool who may be difficult to reach [35], allowing the topic to be explored in more depth amongst each participant who has fundamental experiences to share [36]. Focus groups were conducted online, using Zoom video conferencing. Virtual focus groups offer flexibility and accessibility to participants engaging in discussion over the internet [37].

### Study population and recruitment

To be eligible for the study, participants needed to be at least 18 years or older, speak English, have the responsibility for making decisions about medication for a child or children aged between 0-12yrs, and be living in a rural or remote area in the Australian Northern Territory. Caregivers of children older than 12yrs were excluded. This inclusion/exclusion criteria was applied because children 0-12yrs of age are unlikely to contribute to health-related decisions about medication use.

We purposively recruited caregivers living in or around the surrounding areas of small towns in the Northern Territory. These locations were chosen as most towns outside of Darwin, the capital of the Northern Territory, are considered remote or very remote [38] and people living in remote areas of Australia experience reduced availability of medical services [39]. Participants were recruited using two strategies: social media advertising; and promoting the study via early childhood education and care services. A social media advertisement using the Charles Darwin University College of Health and Human Sciences Facebook account was developed to upload on public Facebook groups and community pages, relevant to the geographical locations. Facebook was utilised to reach people remotely who could view the advertisement and choose to share it with others in their network. Organised playgroups and parent resource centres were also contacted to ask for support with sharing the advertisement on their social media pages. Caregivers who viewed the advertisement and were interested in

taking part, were directed to contact the research team for further details. Eligibility was confirmed using screening questions to check relevant socio-demographic information.

Additionally, using the public register of early childhood education services, primary schools and childcare centres were contacted to request assistance with disseminating the study flyer on parent portals. No exclusions on centres or schools were applied as the research endeavoured to recruit participants from a range of socio-cultural backgrounds representative of the regions. Those organisations who agreed to assist, displayed the study advertisement on parent notice boards, in newsletters, or on their Facebook page. Of the 38 organisations contacted, 21% (8) of those approached responded to offer assistance. Recruitment commenced in March 2022 and continued until July 2022. Participant recruitment resulted in 25 expressions of interest from one region in remote Northern Territory. A total of 15 parents agreed to take part in the study.

## Data collection

Focus groups were facilitated by a female psychologist/researcher (SM) with experience in health-psychology interviewing. Nobody else, other than the research participants (and in some instances their children) were present during sessions. A semi-structured focus group guide, informed by the TPB, was used to ask open-ended questions. Questions related to parental knowledge and attitudes about antibiotic use, antibiotic resistance, sources of information, antibiotic practices, and the perceived accessibility of healthcare (S1 File). The question guide was pilot tested with 3 parents from the researcher's personal network to assess clarity and any issues affecting response prior to the focus groups. Pilot interview data was not included in the analysis.

Focus groups were not stratified by any participant variables, and parents were allocated to a session based on their availability. Every effort was made to support attendance. However, one participant did not attend a scheduled session without advising the research team, and two were no longer able to attend a scheduled group or alternative session. Time required for each focus group was between 45 and 60 minutes when saturation of information was reached. Focus groups were recorded using the Zoom audio-visual recording function. Any observations were noted following sessions. Digital recordings were transcribed verbatim by SM within 7 days following the date of each focus group. Transcripts were de-identified by assigning participants with a study code (P1, FG1 = Participant 1, Focus Group 1 etc) and removing identifying information, including the exact location to protect the anonymity of participants living in small communities. Saturation, when further data collection is likely to yield no new information and additional sampling is considered unnecessary [40], was achieved after the third focus group, consistent with previous observations [41]. All research data was stored securely in accordance with the Data Management Plan approved by the ethics committee. Each participant received a A$50.00 gift voucher after data collection finished.

## Analysis

Transcripts were analysed using thematic analysis. This approach was considered most suitable, as the analytic process can be conducted abductively (in both an inductive and deductive way) [42] to provide a detailed and nuanced account of the data. This was necessary given the study was both theory-informed and exploratory. The 6-stage thematic analysis framework by Braun & Clarke (2006) was used to analyse the data [43]. Data analysis involved the participation of three researchers (SM, MB and SP), and has been outlined in (S2 File) to increase transparency and adherence to proposed standards of data trustworthiness [44].

**Table 1. Participant characteristics.**

| Participant | Number of children | Age of child/children |
|---|---|---|
| P1, FG1 | 3 | 8 months, 6 and 7yrs |
| P2, FG1 | 2 | 7 months and 4yrs |
| P3, FG1 | 1 | 6 months |
| P4, FG2 | 2 | 3 and 5yrs |
| P5, FG2 | 2 | 8 months and 4yrs |
| P6, FG2 | 3 | 3, 5 and 10yrs |
| P8, FG3 | 2 | 4 months and 4yrs |
| P9, FG3 | 2 | 10 months and 2.5yrs |
| P10, FG3 | 1 | 9 months |
| P11, FG3 | 1 | 9 months |

**Note:** P7, FG2 experienced internet disruption and was reassigned to focus group 3.

## Ethics

All participants provided written informed consent after being provided with detailed study information outlined in the Participant Information Sheet. This research was approved by Charles Darwin University Human Research Ethics Committee on 27 January 2022 [reference: H21091].

## Results

Three focus groups were conducted during May, June and July 2022: Focus Group 1 (3 participants); Focus Group 2 (3 participants); Focus Group 3 (4 participants) for a total of 10 participants. All participants were mothers who had between 1 and 3 children. Table 1 describes the variation in participant demographics.

Four themes were generated consisting of nine sub-themes explaining the perspectives of parents and factors guiding mothers when making decisions about their children's antibiotic use. A summary of the themes and corresponding sub-themes is outlined in Table 2.

### Theme 1: Parental knowledge, attitudes and decision making

Participant responses regarding their understanding of antibiotics, their opinions with respect to antibiotic use and the sources which influence parent decisions were labelled 'Parental

**Table 2. Themes and sub-themes.**

| Themes | Sub-themes |
|---|---|
| **1. Parental knowledge, attitudes and decision making** | Knowledge |
| | Attitudes |
| | Seeking advice and reassurance |
| **2. Perceptions of AMR** | Limited understanding of antibiotic resistance |
| | Feelings of complacency |
| **3. Healthcare challenges** | Extended GP waiting periods |
| | Inconsistency of provider |
| **4. Behaviours with antibiotics** | Obtaining antibiotics |
| | Use of antibiotics |

knowledge, attitudes and decision making'. This theme comprises of three key sub-themes: Knowledge; Attitudes; Seeking advice and reassurance.

**Knowledge.** Most mothers were knowledgeable about the indications for antibiotic use. In all groups, mothers talked about the type of infections that should be treated with antibiotics and those infections, which should not: *"When you have a bacterial infection. So, antibiotics aren't going to work with a viral infection. So, um like if you have tonsillitis versus just a sore throat that has no gross bacteria on it"* (P1, FG1).

Mothers responded with some awareness of the effects of using antibiotics, such as the impact antibiotics can have on gastrointestinal health. For example, one mother said: *". . .gut health. I'm not even sure if that's connected to be honest, but that's my perception that antibiotics kind of mess with that good gut health"* (P10, FG3). Other parents were aware of and talked about the effectiveness of the antibiotic being impacted by the child not having the full dose, as a result of medication refusal.

**Attitudes.** There was a tendency amongst participants to associate antibiotic use with recovery: *"Um, I would think it's got a good, um to shorten the sickness time. With, that's what I would be hoping if I'm giving antibiotics and to give their immune system a bit of a, a help to fight illnesses"* (P5, FG2). Most mothers also perceived antibiotics to treat and safeguard against serious illness. For example, one mother described the benefits of antibiotic use as: *"Preventing sickness or something more fatal"* (P10, FG3). Another participant spoke of antibiotic use in relation to treating more severe illness that might deteriorate further, she said: *"Well, I suppose, like if it's. . .a more serious illness that could like get worse, I suppose"* (P3, FG1).

Whilst parents demonstrated some awareness about when antibiotics should be used, their attitudes and beliefs also shaped intentions about the use of antibiotics. The duration of illness, such as a lack of symptom relief, was most frequently mentioned as a concern and was used as a gauge by some parents of when their child might require antibiotics: *"Oh no, we've gone in needing antibiotics. And I think that's where we kind of got how the first child got so sick, was because the GPs here just said 'well, every kid in [town name] has a cough' and you're like, well, they can't, they shouldn't have a cough and it shouldn't go, the respiratory specialist said kids should never have a cough for more than two weeks. If it goes longer than two weeks, there's something else going on that needs to be sorted out, so. Yeah, I think sometimes there is a big reluctance on the side of the GPs to prescribe antibiotics when it is something that does need to be treated by antibiotics"* (P6, FG2).

The nature of illness also appeared to influence some mother's attitudes about the need for antibiotics. Ear infections were most often mentioned as an infection associated with antibiotic use. For example, one mother said: *"Honestly, antibiotics are a lifesaver with an ear infection or something. So, yep, everyone's a bit happier when they're not so sick"* (P1, FG1). Some mothers suggested that pain and fever might be indications for antibiotic therapy and were signs of when to consult the health system for medical advice. There was a proclivity for parents to be risk averse when faced with the situation of managing an unwell child and when symptoms were perceived to be serious or prolonged: *". . .I think I've become quite attuned to when, with my kids. . .it's got to a point that a) they need to see a doctor about it, or I need to speak to one. Um and b) that this is just not going to be fixed with time. I think you know, particularly with ear infections, um you know when your kids in a severe amount of pain when they've got a fever and you know. . .That's when I go, ok, this needs something a bit more than just Panadol"* (P1, FG1). Whilst another parent said: *". . .[he's/she's] had a viral infection and we just needed to make sure it wasn't bacterial because it was day four of fevers and I was like, yeah, this is not ok. . .so you do feel, if you get to that point, it's really because you're desperate to get something or talk to somebody that can give you some reassurance that it's right, or it's not right. . ."* (P11, FG3).

The child's age appeared to influence parental views about the use of antibiotics. Parents were inclined to express concerns about giving very young children antibiotics. This was mostly related to hesitation about adverse reactions and possible effects of the medication on the baby's development: *". . .So, I think that in a really urgent situation where a child is quite ill, yes, definitely antibiotics are the way. . .Um, but other than that I feel that. . .like when they're really, really little is it a good idea to be giving them something that's quite strong, and clearly when they're developing, you know like there's so many other things that we don't give them, you know honey for example. . .But you know like antibiotics are quite intense as well. So yeah, I feel. . .maybe up until one, maybe even two like just when they're absolutely necessary. . ."* (P2, FG1). Another parent said: *". . .especially in babies and young children I worry about side effects on their development"* (P9, FG3).

**Seeking advice and reassurance.** When making health decisions about the use of antibiotics, participants reported they obtained guidance from others. This sub-theme consisted of the sources of information parents often rely on for advice and comfort about their children's health and about antibiotic use. All mothers reported they foremost consider the doctor or prescribers' advice when deciding on antibiotic treatment for their child: *"Um the GP, I suppose, or the health professional that's prescribing them. I trust their judgment"* (P8, FG3). However, parents valued input from others in their social network, such friends and family when making health-related choices and decisions about their children's antibiotic use. Whilst they may obtain this advice differently and for varying reasons, nearly all parents reported to seek additional direction from others to alleviate their concerns. Advice from health professionals in their immediate network who were perceived to be more knowledgeable was highly regarded by many mothers. This often included seeking advice from nursing professionals or those from a health background: *"Yeah, I would listen to the advice of medical professionals pretty well. Ah I still like to have a little bit of my own information and not saying I Google search things. I do it in the right way. And I also have a couple of friends who I would also consult who are from a medical background as well, um and I still trust my gut a lot though, like, I really do. . ."* (P11, FG3).

Participants also reported they might seek advice from assumed health mavens about antibiotics. These were people perceived to be more experienced, such as friends or family with older children who may have encountered a similar health situation with their own child in the past: *". . .potentially like I've got some like obviously being a first-time mum like I, I've got other friends that have some older kids as well, so like they've been parents and um, maybe like seek a bit of their advice as well"* (P3, FG1).

The way in which information was obtained from others in the social network varied, and parents may seek advice from those that they know through mother's groups, social media or by contacting friends and family directly. Mothers responded that they may also undertake their own research, such as searching the internet for antibiotic information and other's experiences to inform their decisions about the most optimal antibiotic and flavour when consulting the doctor. The reasons discussed by mothers for engaging others in the network varied, and the following scenarios were mentioned: for general advice about antibiotic use; for signs of improvement when the child is taking antibiotics; for tips about administering antibiotics; for guidance with foods the child can consume while taking antibiotics; for reassurance about antibiotic side effects; for advice about the child's illness when difficulties were encountered accessing a GP; and when to consult the health system. For example, one mother spoke about seeking guidance from her friends regarding when to consult a doctor: *"I usually go to my friends prior to seeking medical, like official medical channels. Um, just to see if I'm being an overprotective mum or ah, sometimes I think I'm a bit slack and I just think, they'll be ok. And so, I always like a second opinion to make sure that I'm not like sending my kids down the sepsis*

*pathway and, um, yeah making sure I'm seeking treatment at the right time, and to alleviate my own concerns. And often in the middle of the night rather than heading up to ED, I might text some friends who I know will be awake and ask their advice and double check, make sure. Yeah, because yeah, rather than having a sick baby who's worse in the morning, I can yeah, alleviate my own stress that way*" (P5, FG2).

### Theme 2: Perceptions of AMR

Parent responses with respect to how they perceived and interpreted antibiotic resistance were categorised as 'Perceptions of AMR'. This theme covers the response patterns about bacterial resistance and includes two key sub-themes: Limited understanding of antibiotic resistance and Feelings of complacency.

**Limited understanding of antibiotic resistance.** All mothers in the sample had heard of antibiotic resistance or super bugs, and there was some awareness and concern of the relationship between antibiotic use and resistance: "*Well, I haven't got lots and lots of research, but I do know that there's some resistance to antibiotics. Um, that sort of plays a little bit in my mind, but once again I'm not super-duper informed on it. So, I wouldn't be making a judgment just on that, but that is something that, you know, if it came up and there was many situations where I had to have [him/her] on antibiotics. Then you would probably look into it a little bit more, and what type of antibiotics they're using. . .*" (P11, FG3).

However, the problem of antibiotic resistance was not well understood, and parents weren't informed about how AMR functions. Half the participants associated the occurrence of antibiotic resistance with the individual not finishing the treatment course: "*Hm, I've heard of super-bugs, I don't know much about it. Um, and resistance I guess I just, my information with antibiotics is very limited. I just know you have to take the whole course for it to work, otherwise it could be extremely bad*" (P10, FG3). Participants did not generally articulate their awareness of further implications of the development of bacterial resistance, such as the transfer of bacteria to others.

Whilst understanding of how bacterial resistance develops and is facilitated was limited, parents reported some concerns about antibiotic use and the potential for antibiotic resistance when giving their children antibiotics. Although there was some understanding amongst participants that antibiotic resistance occurs at the individual level, a connection wasn't articulated that AMR is also a societal issue in which resistant bacteria can spread between people and from animals to people, and can affect anyone, at any time. Many participants attributed only the individual's antibiotic use, with some mothers perceiving that overuse as a child may impact the body from responding to antibiotics in later life. Parents didn't recognise that the behaviour of others would necessarily impact whether they were a carrier of resistant bacteria, and their response to antibiotics. Descriptions that the person changes in response to antibiotics, rather than the bacteria were noted: "*. . .I don't want the kids in the future, you know, like 50 years down the line if they do get something, it's just like, oh well, you had a lot of antibiotics as a kid um, and you know we're not going to be able to treat this. . .*" (P2, FG1).

**Feelings of complacency.** There was a sense of complacency amongst other parents about the potential for AMR, and the likelihood of being personally impacted was viewed as a distant risk more pertinent to the hospital system. Perceptions that AMR is less relevant to Australia because prescriptions are required to obtain antibiotics, and views that antibiotics are not over-prescribed in Australian children were expressed: "*. . .I think that what I know about it is that it might have emerged more from countries where things, where antibiotics are available over the counter, as opposed to being prescribed and being an actual medical plan, where in other countries it's overuse is quite clear. . .I think my experience of medical professionals in here, has*

*been that they are actually very, they don't prescribe antibiotics very easily, particularly to children. . ."* (P4, FG2).

### Theme 3: Healthcare challenges

Participant views with respect to how they perceived the accessibility of healthcare when their children are unwell, were classified as 'Healthcare Challenges'. This theme consists of the commonalities shared by parents with regards to the difficulties encountered when accessing the health system. Two sub-themes were constructed, which highlighted the barriers parents spoke of when seeking appropriate medical care for their children: Extended GP waiting periods and Inconsistency of provider.

**Extended GP waiting periods.**   Visiting the GP is usually the first encounter with the medical system when a patient is unwell. Nearly all mothers raised difficulties with their children being seen by a GP in a timely manner. Wait times for GP appointments was the predominant healthcare challenge noted by mothers in all focus groups: *". . .it is really hard to get into the doctor, you kind of need to know like ahead of time that you're going be sick and book in a week in advance"* (P6, FG2). Experiences of waiting periods varied between participants from 1–4 weeks, which depended on whether parents were prepared to visit any doctor or wanted their child to be reviewed by their preferred GP. Half the sample expressed that the difficulties of seeing a GP had been further exacerbated since the introduction of COVID-19. Some parents agreed that they tended to consult people from their personal network for advice regarding their child's health during waiting periods to see a GP: *". . .it takes so long to get into the GP, in that like waiting period when you're like, what am I doing? what could this be?"* (P8, FG3)

Interviewer: Worrying?

*"Yeah, in the worrying phase of like, hm you know, that's when you kind of turn to friends. Even just to like whinge"* (P8, FG3).

Most parents reported that difficulties accessing a GP appointment in a timely manner may result in decisions to attend the local hospital Emergency Department (ED). Mothers acknowledged they didn't want to utilise the hospital system, but if they were concerned and perceived a medical review was necessary, they considered this a safety net option: *". . .I think it's hard because sometimes it gets to a point where you're trying not to overreact to something. So, by the time you, you actually do make an appointment, you kind of need one within like the next couple of days at least, if there's something that's going on like that, you know, if you suspect your child has a bacterial infection, you don't want to wait ten days before you can get some treatment for it. So, um, yeah, they've been really good at ED. . ."* (P11, FG3). Another parent explained: *". . .if one of my kids was very sick and I could not get an appointment then I would take them down to the hospital. Although I know that. . .it's not a GP service and I know you should only be going there if it's an emergency. But to be honest, it is an emergency if you can't get antibiotics or, or get your kids seen to in a timely manner and they're in pain and you're concerned for them. . ."* (P1, FG1).

**Inconsistency of provider.**   In all focus groups mothers raised issues about a lack of continuity of care, explaining that GPs in their town are frequently booked out in advance, resulting in there being low availability for the preferred doctor. This resulted in parents consulting other GPs available at the same practice or seeking appointments at a different practice in order to have their child reviewed. At times this resulted in parents being dissatisfied with the outcome of the consultation. Scheduling a medical appointment ahead of time was considered

the only way of consulting the same GP, as one mother explained: *". . .unless you book in advance, you will very rarely see the same doctor"* (P4, FG2).

## Theme 4: Behaviours with antibiotics

Participant responses with regards to how they used antibiotics were labelled 'Behaviours with antibiotics'. The themes that emerged broadly reflected parental expectations for receiving prescriptions, how well antibiotics were used when prescribed and what parents did with antibiotics after they discontinued using them. Two key sub-themes were generated: Obtaining antibiotics and Use of antibiotics.

**Obtaining antibiotics.** Patient expectations can influence the prescribing behaviour of medical stewards. Half the sample discussed that there had been times when they knew their child required antibiotics prior to consulting the health system. While most mothers responded they had not attended a medical appointment expecting to receive antibiotics and were seeking an examination for their child, 40% of parents reported they had consulted a doctor intending to receive a prescription for antibiotics. Some mothers discussed conducting research prior to the medical appointment, which informed their understanding of the treatment requirements, or prior experience guided decisions about the preferred type of antibiotic to be prescribed: *"I have gone into doctors knowing that the condition that my child had required antibiotics. . .where I have looked it up or you know this is the second time, they've had it, or my first, first one had it and now my second one has it, or um, and I know that's the treatment for it. Then I would go in with the expectation to walk out with an antibiotic. And sometimes the type of antibiotics as well. So, I know that antibiotic cream might work better for something from experience, and I would push for that instead of oral antibiotic. Things like that"* (P4, FG2).

Another parent responded that she had also experienced times when she preferred to receive a prescription for antibiotics, said she: *". . .sometimes I've had prescriptions and I haven't filled them straight away because I've just wanted to hold off a little bit longer, but have that option there just if I feel like my kid needs it"* (P5, FG2). The interviewer asked the mother why she thought that antibiotics might be necessary. The mother explained: *". . .If my child's not improving, ah, if they seem to be worsening, if they um, their needing a little bit of help, their immune system seems to not be doing it on their own. Yeah"* (P5, FG2). A mother in a different focus group discussed her beliefs about when her children required antibiotics and her intention to receive a prescription when consulting the doctor: *"Pretty much if I'm taking them to the doctor, it's because I know they need antibiotics. If not, I'm not taking them to any germ-infested doctors' surgery or hospital unnecessarily. So, like for example my [child] recently, I'm pretty sure [he/she] had a [medical condition]. . .And I didn't even need to go and see the doctor. So, I have a good relationship with [him/her], and I literally just sent [him/her] an email and I said this is what's happening. Can I avoid coming in with you? Because going anywhere with one kid, let alone two or three to a doctor's surgery, is not my idea of a good time. . .so [he/she] said yep, sounds like a [medical condition], I've written the prescription, I'm going to drop it to your house, and the bus dropped it off. . ."* (P1, FG1).

**Use of antibiotics.** Parents responded that when their children were prescribed antibiotics, 60% didn't always follow the medication instructions precisely. The most commonly reported issue affecting medication adherence was prematurely terminating the child's antibiotic course, followed by missing doses due to forgetting. Some of the reasons described by mothers for non-adherence were unavoidable and highlight the challenges of administering antibiotics successfully to children. The reasons varied for stopping antibiotics early and were influenced by factors such as: side effects; perceived lack of improvement; child refusal: *". . .so,*

*I was feeding [him/her] every hour and [he/she] was losing weight. Then I asked for a different antibiotic, which [he/she] improved slightly. But [he/she] wasn't having the gains that a baby should have. . .And then that's when I said to [him/her], I would like to stop. . ."* (P2, FG1). Another parent said: *". . .I just saw it wasn't working and like there was literally no change. So, that would be one reason for stopping one type of antibiotic and actively asking for something else to make a difference, which did work. . .we have on occasion, when it's been really difficult, asked for a different type and for like flavour wise and stuff. . ."* (P1, FG1). When the antibiotic was discontinued due to perceptions of minimal improvement, issues with palatability or side effects, a change in antibiotic was usually requested by the parent so that treatment could resume. However, child symptom relief also influenced parental decisions to not fill a repeat or not complete final doses of antibiotics. In these instances, seeking a repeat or purchasing further medication that had run out was regarded as unnecessary by the parent: *". . .I think in the past I've not filled a repeat maybe, or instead of giving it for seven days, the antibiotic has run out after six days and I've not bothered to complete the last day because my child's better. . .I try and always finish the course. I don't stop giving if I've got the antibiotics, but I wouldn't fill a repeat for instance, if the kids improved and we've run out. I wouldn't bother seeking more anti-biotics"* (P5, FG2).

Most mothers responded they would not keep leftover antibiotics for reuse, and parents expressed concerns about reusing antibiotics with their children. Half the sample mentioned discarding leftover antibiotics in household waste. Some parents were hesitant of the risks of reusing antibiotics with their children but were more open to this practice for themselves: *"No. If it's a tablet, like if it was for an adult, I think it's a different story and you can see the use-by date on it and like I'd be quite happy to take that myself in six months' time if I felt like it was the same sort of thing. I'd be like yep, alright I'm going to give this a go and avoid going to the doctors, but for kids, no. . ."* (P1, FG1).

Other parents perceived that antibiotic fluids expire too quickly to be retained and while creams were perceived as more likely to have residual value. A small number of mothers shared that they have or may keep antibiotic creams or ointments for later use: *". . .Once I think when my [child] had [medical condition] and we were like well, the younger one is defi-nitely going to get them. . .we might have kept the, a tube of, you know cream, antibiotic cream, just in case [he/she] got them, and it would be the same presentation. It would be the same. So, I would feel confident then to use it. But otherwise, no"* (P4, FG2). One mother reported to always keep leftover antibiotics and would consider reusing them if the same medication was re-pre-scribed or the illness recurred to save money: *"I always keep them, but I don't use them. . .and then I throw them out once their five years expired. . .but I don't think I've ever actually been re prescribed the same thing while I've had them, so I've never used them. If it was maybe within a few months and it was a recurring illness, I'd probably keep them just to save buying the exact same thing. But I've never had that experience, but I might, I might reuse them"* (P5, FG2). Another parent spoke about her apprehension to use leftovers. However, she discussed that if antibiotics were reused, the decision would involve input from respected people in her net-work, she said: *"No, I don't think when it comes to my baby that I would risk that at all. . .I hope-fully wouldn't have any left. I haven't been in a position, but if there was something there um, and a very respected friend of mine who is a very straightforward kind of person and [he/she] said, honestly you need to do this if you've got some, start it. . .I possibly might consider it. But not on my own whim or my own thoughts. I would never pull them out of the cupboard and be like, yeah, here we go"* (P11, FG2).

Nearly all mothers expressed strong beliefs they would not share their children's antibiotics with the children of others and were aware of the risks involved in this practice. Parents were concerned about harm to the other child, such as the child receiving the incorrect dosage or

allergic reactions: *"No. No, I'm not killing somebody else's child by accident because I want to play doctor, definitely not. No way"* (P2, FG1). Some parents raised that their child needs the medication first, which has been prescribed for them, and if using leftovers, the other child would receive an incomplete course, or there would be insufficient left to share. Mothers perceived that other parents should seek appropriate medical channels for their children when using antibiotics. One parent disagreed and expressed her thoughts about sharing antibiotics and being open to this practice: *"I don't think I would take someone else's antibiotics for my child, but if for instance one of our close playmates got sick and I had some leftovers. I'd quite happily. . .give it to [his/her] parents and they can make that decision. I don't think I'd do it for my kid, um, but yeah"* (P5, FG2).

Interviewer: And what do you think would influence your decision to share do you think?

*"Um probably money. I'd think that'll save a trip to the doctor. That'll save you buying some. I just, if I had some leftovers, I'd be happy to share. It's a bit of a waste otherwise, yeah. . .Like and sort of sharing community love. We share our illnesses we might as well share our cures"* (P5, FG2). Another participant in the same focus group responded suggesting that decisions to share antibiotics might be influenced by a lack of bulk billing services: *"So too, and also on the back of what P5, FG2 said, [the town] doesn't have a bulk billing clinic. There's no bulk billing doctors in [town name]"* (P4, FG2).

## Discussion

This study explored factors that influence the decisions of parents living in remote areas about the use of antibiotics in their children. The first research question investigated was *'Does knowledge and beliefs about antibiotics influence parental attitudes regarding the use of antibiotics with their children?'* In this research we found evidence that most mothers displayed some accurate knowledge that antibiotics treat bacterial infections, which contributed to their general understanding of the role of antibiotic medicines. However, decisions about the use of antibiotics were not always knowledge based, and parental beliefs also guided opinions and behaviour. Choices to consult the health system and beliefs about when antibiotics might be necessary were often driven by fear of serious illness. The perceived seriousness of the child's illness was an indication for the need for antibiotics, irrespective of the cause of the illness (known or unknown). Perceived signs of severe illness included the child's level of discomfort, the nature of illness, duration, and a lack of improvement. At times, parents associated these signs as possible indicators of bacterial infection. Parents believed antibiotics would promote recovery by shortening the duration of illness and safeguard against further deterioration. This resulted in some parents actively seeking antibiotic prescriptions for their child from the doctor, which provided parents with relief from worry about possible adverse health consequences. Although some awareness about antibiotic use and resistance was expressed, parents had limited understanding of how AMR functions and there were misconceptions about the broader implications of the problem. When managing an illness threat, parents tended to direct their focus towards their child's immediate wellbeing, which appeared to supersede concerns of AMR. However, parent propensity to be risk aversive with their children was influenced by the age of the child. Most mothers were cautious about the impact of antibiotics and possible side effects on a young child's development and mothers were more reticent towards using antibiotics in newborns and infants unless it was essential.

While many international studies have identified parental misconceptions about the intended use of antibiotics and inappropriate behaviours towards using antibiotics [2–5], the parents in this study were generally aware that antibiotics are not used to treat viral infections. However, beliefs about illness severity and antibiotic use mirrored previous international

studies amongst parents [26]. Our findings are similar to qualitative UK research where the intention to seek antibiotics was influenced by maternal worry and mother's trust in antibiotics and their beliefs that antibiotics symbolise recovery, healing and safety [29]. Qualitative international research also supports our finding that mothers have concerns about antibiotic use in very young children [14], though parents relax their concerns about the risk of antibiotics as the child became older [13]. Consistent with previous Australian research, some parents in this study believed antibiotics are required to treat ear infections and thus hold assumptions incongruent with guidelines for managing acute otitis media [30]. Behaviours are also influenced by the difficulties parents may experience with watchful waiting when their child is distressed [24].

Findings in this study that beliefs influenced behaviour are consistent with the TPB model used to inform this research. In particular, our results highlight the role beliefs can play in guiding parental attitudes and decisions with respect to their children's antibiotic use. Even when parents have existing knowledge about the appropriate use of antibiotics, underlying beliefs about using antibiotics may override this understanding and govern decisions. As evident in this study, earlier life experiences including the history of the individual, the context and emotions can influence formation of the belief system and drive attitudes and behaviours towards the use of antibiotics.

The second research question explored in this study was *'Does limited access to primary health services lead to the use of social networks for medical advice regarding child antibiotic use?'* This research identified that limited access to healthcare, such as long waiting periods to access a GP appointment, resulted in some parents obtaining direction from respected people in their network regarding their children's health needs in the interim. Whilst advice from the doctor was preferred, timely access to a GP wasn't readily available to parents living remotely. Mothers in this study accessed the support of friends and family in both professional and non-professional health roles for guidance about their decisions. This broadly related to seeking reassurance to alleviate concerns, and suggests that a driver of parental behaviour, including obtaining advice from both medical and non-medical contacts, is fear. In addition, parents reported to undertake their own research to find illness information and advice about antibiotics. Information from others was taken into consideration for a range of factors related to their children's antibiotic use. Parents may also act on the advice of people in their social network about when to consult the health system and for antibiotic-related decisions, such as decisions about using leftovers. This is consistent with international study findings where parents who live in resource limited settings obtain advice about antibiotics from friends and relatives [18, 22].

Consistent with the TPB model, parent choices were influenced by subjective norms. Parents in this study sought guidance about consulting with GPs and the use of antibiotics from friends and family, and this information was both valued and factored into decisions. Findings indicate that the decision-making of mothers living in rural and remote communities, involves talking to others and having networks available to obtain advice, notably when access to healthcare is difficult. Subjective norms might also contribute to the decisions of urban parents regarding the use of antibiotics, which needs to be investigated in future research.

The third research question examined in this study was *'Does limited access to primary health services contribute to parental decisions to pressure GPs for antibiotic prescriptions?'* Parents in this study did not report that they place demands on prescribers for antibiotics. However, in somewhat of a contradiction, parents reported actively seeking antibiotic prescriptions with 40% of the sample responding that they had attended a medical appointment intending to receive an antibiotic prescription for their child. The need for antibiotics was influenced by past prescribing behaviour and the parent gauging their child's symptoms.

Parents discussed issues of GP waiting periods, utilisation of emergency services and low continuity of care due to doctors in the region having limited availability, which impacted their children seeing a regular GP. At times, this resulted in parents being dissatisfied with seeing an alternative doctor with whom they may not have the same level of trust as they do with their preferred GP. However, mothers did not relate these healthcare challenges to their reasons for seeking antibiotic prescriptions.

Numerous international studies have documented parent requests for antibiotics from prescribers [1, 5, 7, 8], including in low-resource settings [9]. While Australian researchers have reported parents living in metropolitan areas do not expect GPs to prescribe antibiotics for their children [31], our rural sample often expected such prescriptions. A possible explanation relates to the reduced accessibility of health care in rural and remote environments such that there may be a greater degree of parental initiative in Australian rural populations and an enhanced attitude of support for independent decisions about antibiotic use.

The TPB model can be used to understand parent intentions to obtain antibiotics from the doctor. We found that parents hold the attitude that seeking antibiotics from prescribers is an acceptable behaviour and that parents perceive behavioural control in receiving antibiotics when desired in that doctors will acquiesce to their wishes. Parents who intended to receive antibiotics entered medical consultations with pre-established treatment expectations, which influenced the provision of prescriptions, and in some instances, parents guided the doctor about the type of antibiotic to be prescribed.

The final research question explored in this study was *'Does limited access to primary health services influence parental decisions to store/share antibiotics?'* Most mothers reported they do not store or share antibiotics. However, when antibiotic leftovers were retained, decisions were influenced by convenience, the type of product and cost saving strategies to potentially re-treat a similar illness in the child or sibling. Intention to share antibiotics was driven by having leftovers and the opinion that sharing would assist friends to save time and money. In this study, we found evidence to support that limited access to bulk billing services (free health care) can influence decisions to store and share antibiotics. The cost incurred by parents to consult a GP, purchase antibiotics from the pharmacy and the time and effort required to engage with health services was factored into decisions to store and intentions to reuse and share leftovers.

Our findings are consistent with international studies that parents may use leftover antibiotics to treat alike or reoccurring symptoms in their children [7, 22] and reuse antibiotics between siblings/other children [4, 8]. Use of non-prescription antibiotics in children has also been associated with economic constraints [1, 17] and poor access to healthcare [16, 18–20]. Our findings indicate that keeping leftovers is problematic and can promote reuse and intentions to share by having antibiotics on hand when illnesses present. International research highlights the significant association between retaining leftovers and parent choices to autonomously use antibiotics [6].

Attitudes about the appropriateness or inappropriateness of storing leftovers and sharing antibiotics were influenced by restricted access to health services. In this instance, and aligned with the TPB, behavioural intentions to retain leftover antibiotics for future use and/or to share residual antibiotics, which is considered an *inappropriate* antibiotic use behaviour, was related to a lack of perceived behavioural control to act appropriately. Specifically, being unable to consult with a prescriber either in a timely or affordable manner impacted upon intentions with respect to the use of antibiotics. Conversely, storing leftovers increased perceived behavioural control over healthcare barriers by having access to antibiotics. This influenced intentions to reuse antibiotics to treat similar symptoms or to share leftovers to assist others.

This study found differences in findings between first time mothers of a child less than 12 months of age and mothers with more than one child. First time mothers with babies had the most optimal behaviours with antibiotics and did not report intentions to receive antibiotic prescriptions, medication adherence issues, storing or sharing behaviours. This finding was likely due to these parents having less exposure to using antibiotics and managing childhood illnesses than parents with older and multiple children. It may also be related to the finding that parents are cautious in their approach towards antibiotic use in infants. Findings from this research suggest that *both* optimal and inappropriate behaviours with antibiotics were driven by fear. Mothers appeared to be cautious about the vulnerability of very young children to a foreign substance and were reserved about using antibiotics. Furthermore, concerns about non-prescription antibiotic use were driven by fear of the child experiencing an adverse event. Conversely, antibiotic seeking behaviour was influenced by maternal fear of serious illness. One observation was that mothers were risk averse when managing an illness, particularly when it was perceived to be serious. This appeared to influence their judgement about the anticipated benefits of using antibiotics and the perceived risks of use were diminished. Health service delays and reduced access to a regular GP contributed to parents being more self-reliant and seeing what else they can find out from other trusted sources. Difficulty accessing health services supported a culture of self-informed medical decisions and expectations about the provision of antibiotic prescriptions. The likely difficulty of obtaining a follow up appointment also influenced mother's intentions to obtain antibiotics from the doctor.

Our findings indicate that mothers in rural and remote areas who can't get access to accessible health services make decisions about antibiotic use out of fear and based on the advice of other sources when they perceive their child is vulnerable to a health threat. This basic parental drive to nurture their children can be used to encourage appropriate behaviours with antibiotics by supporting parents to understand the risks of use across the age range, as well as the harmfulness of self-directed practices. Based on our findings, knowledge of appropriate indications for antibiotic use is insufficient to ensure appropriate antibiotic use behaviours. Enhancing risk perception about the potential effects of antibiotic use and misuse (across the child age span) may be more successful at quelling inappropriate behaviours by reshaping risk awareness and re-framing underlying beliefs about antibiotics. This study found that parental decisions about the use of antibiotics conforms to predictors associated with the TPB. As such, future interventions should specifically target these predictors, in particular, the influence of social norms where ready access to health care advice is unavailable. This might include an evaluation of the use of social media as an educational platform or the development of bespoke web-based information and decisional-balance tools to assist parental decision making about the appropriate use of antibiotics. While patient behaviour is a significant contributor to antibiotic use and misuse, and thus to the development of AMR, this study highlights contextual variables (such as geographical location) that may also affect prescriber behaviour. Future research should address the impact of rural and remote locations on the decisions of prescribers and the potential contribution of these decisions to the AMR conundrum.

## Study limitations

We would have liked to recruit more caregivers living greater distances from towns and from a variety of regions. Distance education providers were contacted about promoting the study to assist with reaching families living in the most remote parts of the Northern Territory, including indigenous communities, cattle and mining stations. However, the participants in this study were all females residing in or near a larger town who possessed enough computer literacy skills to engage in an online focus group. The requirement of participants to have internet

access and computer skills may have generated sample bias towards a higher level of education and socio-economic status amongst participants. However, online discussion also offered flexibility to parents to engage in discussion over the internet during coronavirus disease 19 (COVID-19) and improved accessibility for those residing further distances from towns. Variation in antibiotic opinions and behaviours might differ in a sample that included male parents and parents of lower education and/or without access to computer and internet. However, other qualitative studies examining parental views towards antibiotic use have documented low participation from males [27, 28].

Challenges were encountered recruiting participants for this study, resulting in group size limits. We had a relatively small sample, notwithstanding data saturation being reached. A review of 170 focus group discussions found that participant numbers varied from 3 to 21 participants per focus group, highlighting the variability in group sizes [35]. In our study, a low response rate from organisations agreeing to participate in promoting the study contributed to difficulties accessing a larger sample of caregivers from a variety of regions. In the smallest towns, there were fewer early educational services and the response rate from these organisations was particularly low. This lack of responsiveness might reflect a general level of hesitancy amongst members living in small communities to take part. Future studies will need to take into consideration the low response rate of communities in rural and remote areas. These limitations restrict the generalisability of the research findings to the wider population. Further research is needed to recruit larger groups, including fathers, from a broader range of areas to increase sample size and diversity amongst participants.

Focus groups may also encourage participants to give socially desirable responses because of the social dynamic of the interview. However, mothers in our sample appeared to discuss their opinions and experiences candidly. There was a willingness amongst participants to share their views even when they differed from other participant responses and may have been perceived as undesirable. This openness by participants provided rich data to investigate the research questions.

## Conclusion and recommendations

While it was clear that parental decisions about the appropriate use of antibiotics was influenced by the availability of steward advice, the drivers of such behaviours were extant attitudes and beliefs, the social norms of the population, and perceived behavioural control over both access to antibiotics and access to informed antibiotic advice. These drivers were consistent with the TPB health behaviour model. Additionally, this research uncovered additional variables influencing parental behaviours, including risk perception and aversion, age of the child, and knowledge about AMR. The results were broadly consistent with international literature. Findings from this study provide an impetus to further explore the value of the TPB in both predicting antibiotic use behaviours by parents in rural and remote locations using quantitative research. Subsequently, interventions can then be developed that target the specific identified drivers of antibiotic use behaviours within discrete populations, which would offer a valuable contribution to the management of AMR.

## Supporting information

**S1 Checklist. COREQ checklist.**
(PDF)

**S1 File.**
(PDF)

**S2 File.**
(PDF)

## Acknowledgments

We would like to thank the parents who contributed to this research and the organisations that supported the promotion of this study.

## Author Contributions

**Conceptualization:** Stephanie A. Marsh, Mitchell K. Byrne.

**Data curation:** Stephanie A. Marsh, Mitchell K. Byrne.

**Formal analysis:** Stephanie A. Marsh, Mitchell K. Byrne, Sara Parsafar.

**Investigation:** Stephanie A. Marsh.

**Methodology:** Stephanie A. Marsh, Mitchell K. Byrne.

**Project administration:** Stephanie A. Marsh.

**Resources:** Mitchell K. Byrne.

**Supervision:** Mitchell K. Byrne, Sara Parsafar.

**Visualization:** Mitchell K. Byrne.

**Writing – original draft:** Stephanie A. Marsh.

**Writing – review & editing:** Stephanie A. Marsh, Mitchell K. Byrne, Sara Parsafar.

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
