## [Decision Letter · Decision Letter 0]

13 Apr 2023

PONE-D-23-05601What influences parental decisions about antibiotic use with their children: A qualitative study in rural AustraliaPLOS ONE

Dear Dr. Mitchell K Byrne, Ph.D.,

Thank you for submitting your manuscript to PLOS ONE. After careful consideration, we feel that it has merit but does not fully meet PLOS ONE’s publication criteria as it currently stands. Therefore, we invite you to submit a revised version of the manuscript that addresses the points raised during the review process.Please ensure that your decision is justified on PLOS ONE’s publication criteria and not, for example, on novelty or perceived impact.

I agree with the comments from the reviewers. It is good to frame the introduction section to make it better as suggested but not limited to three paragraphs.  

We look forward to receiving your revised manuscript.

Kind regards,

Kshitij Karki, MPH, MA

Academic Editor

PLOS ONE

Journal Requirements:

Additional Editor Comments (if provided):

Thank you authors. The major issues related to antibiotic use among children were highlighted.

Reviewers' comments:

Reviewer's Responses to Questions

**Comments to the Author**

1. Is the manuscript technically sound, and do the data support the conclusions?

Reviewer #1: Yes

Reviewer #2: Yes

2. Has the statistical analysis been performed appropriately and rigorously? 

Reviewer #1: I Don't Know

Reviewer #2: N/A

3. Have the authors made all data underlying the findings in their manuscript fully available?

Reviewer #1: No

Reviewer #2: Yes

4. Is the manuscript presented in an intelligible fashion and written in standard English?

Reviewer #1: Yes

Reviewer #2: Yes

5. Review Comments to the Author

Reviewer #1: Thank you for the opportunity to review this manuscript. The topic is important, and becoming increasingly so. While the authors note there are similar studies in many parts of the world, Australia has a dearth of literature in this area of inquiry.

Background: the authors nicely outline the issues, and actually list in the first paragraph the findings from this study. So assuming the hypothesis was they would find similar outcomes. The theory seems appropriate for the research questions. Methods: The choice of FGDs seems appropriate, however, the overall number of participants was quite small and from one geographic area. Given the FGDs were conducted online, a smaller group per FGD was likely important. It would be interesting for the authors to speculate a bit more on why they had such a poor response rate given the wide outreach during recruitment. Also, is it possible the participants knew one another since they were all from one region? Data Collection: no concerns other than the small number of participants. Results: It would be useful to include the age range of participants in addition to the number and age of their children. Did findings differ among participants based on the number or ages of their children, e.g. 40% of the participants stated they would give antibiotics to a friend, etc etc (just an example - it is still possible to quantify aspects of a qualitative study). Discussion: I would love to see more pointed recommendations for those trying to combat AMR. There were important findings such as the use of the internet, the sharing of medications, etc. It would be helpful to do a f/u study of providers to understand their perspective and experience prescribing antibiotics in this setting.

Overall this was a well-written manuscript. My concerns are primarily with the small number of participants, all from one region. The authors do address limitations, but I do feel this reads more as a pilot study in terms of reaching the population of interest. As an aside, there are several places where copy-editing is needed, e.g. double-checking comma placement

Reviewer #2: Thank you very much for allowing me to review your work. It is a very interesting study on the drivers that induce the use of antibiotics, as well as the knowledge about them in a rural population in Australia. Knowing these drivers will allow us to establish activities and interventions that enable a better use of antibiotics.

This is a very interesting study that can lay the foundation for quantitative studies.

After reviewing the manuscript, I would like to make some comments:

Abstract:

The abstract should end with the implications of the results. Apart from that, the abstract is well-structured and focused.

Introduction:

The introduction is too long and difficult to read. It should be structured into 3 paragraphs. The first paragraph should highlight the background of the research, from a global to a more local perspective. The second paragraph should demonstrate the current state of the research topic, putting it into context with other research results around the world. The last paragraph should highlight what is yet to be known about the research topic and what the main objective of the study is.

Methodology:

It is explained that due to COVID-19, the focus groups were conducted via zoom. I believe this should be discussed as a strength or weakness of the design, rather than using it as a justification for conducting the focus group interviews.

In the data analysis section, was thematic analysis done with any software? In addition, it is explained that 2 researchers conducted the analysis, but did they do it independently? How did they resolve interpretive discrepancies?

In the same section, it should be informed how the focus groups were concluded. Were they ended when information saturation was reached?

Results:

They are well-structured, explanatory, and allow for directing the discussion.

Discussion:

Well-structured. It is very comprehensive.

Conclusions:

It should be shortened. It is too long. It should be summarized in one conclusion. Additionally, it would reinforce that the main implications of the study should be discussed in this section.

6. PLOS authors have the option to publish the peer review history of their article (what does this mean?). If published, this will include your full peer review and any attached files.

Reviewer #1: No

Reviewer #2: **Yes: **Juan M. Vazquez-Lago

---

## [Author Response · Author response to Decision Letter 0]

29 May 2023

RE: Response to Reviewers

29 May 2023

RE: Response letter to reviewers

Dear Dr. Karki,

Thank you for considering our manuscript titled “What influences parental decisions about antibiotic use with their children: A qualitative study in rural Australia” for publication with PlosOne. We have noted and carefully taken into consideration the feedback from the reviewer(s) and yourself. Please see below our responses to each point raised during the peer review process: 

Reviewer 1:

The overall number of participants was quite small and from one geographic area:

• We agree that the overall sample size was smaller than we had planned and that participants were from one geographical area, rather than a range of regions targeted during recruitment. We acknowledge this as a study limitation. However, we do note that saturation was reached after three focus groups, which is consistent with past research and meets minimum sampling requirements. The existing literature suggests that achieving data saturation ensures the trustworthiness of qualitative data analysis. Saturation indicates that all relevant themes and concepts have been adequately explored, and further sampling may not provide any new information. Therefore, saturation can override the importance of sample size in qualitative research. As suggested by Namey et al. (2016, p. 2), "It [Thematic saturation] is a critical point in a qualitative study; identifying when thematic saturation is reached minimizes the likelihood of conducting unnecessary data collection. It is also the most commonly referenced method for estimating qualitative sample sizes (Guest et al., 1995; Sandelowski, 1995)”. Furthermore, we argue that the smaller group sizes appeared to have some advantages over a larger group of parents. We were able to collect quite in-depth information from each participant regarding their perspectives and personal experiences, which is consistent with some of the benefits noted in the literature with using mini-focus groups. Parents were very open and elaborative in their responses indicating a high degree of comfort to share and respond to one another. This might not have occurred in a larger group setting and this breadth of data provided considerable insight into how decisions were made. 

Reference: Namey, E., Guest, G., McKenna, K., & Chen, M. (2016). Evaluating bang for the buck: A cost-effectiveness comparison between individual interviews and focus groups based on thematic saturation levels. The American Journal of Evaluation, 37(3), 425-440. https://doi.org/10.1177/1098214016630406

It would be interesting for the authors to speculate a bit more on why they had such a poor response rate given the wide outreach during recruitment:

• It is likely that the reason participants were from one geographic area was because this town was the largest. The other regions had much smaller population sizes and there were far fewer early educational services and advertising channels to recruit through in these smaller towns. We observed that the organisations, particularly those in the smallest towns, were not responsive to advertising the study, which limited dissemination of recruitment information. We speculate that the overall poor response rate from early education providers might reflect a general level of hesitancy in small communities to participate due to the greater potential for participant identification. We have added two sentences to the paper (page 32, lines 759-762 on the marked-up copy) to elaborate further about the low response rate.

 Is it possible the participants knew one another since they were all from one region? 

• While it is possible that participants may have known one another, there were no indications of this throughout recruitment or data collection. Had they known one another, the possible positive implications for the study would be that familiarity encouraged participants to extend on the responses of others to explore more nuanced insights and social behaviour. However, we reiterate that there was no evidence to suggest that this was the case. Participants also completed a Confidentiality Agreement Form prior to taking part in a session and no ethical issues were raised throughout the research.

It would be useful to include the age range of participants in addition to the number and age of their children:

• While we agree with the reviewer that including participant (parent) age ranges in the results table might have been useful, this information was not collected from participants. The reason we chose not to collect this type of personal information is that we actively sought to limit the number of in-direct identifiers, which in conjunction might identify participants in small communities who have been quoted verbatim. This was an important ethical consideration.

Did findings differ among participants based on the number or ages of their children?

• Differences in findings were observed in the behaviour of first-time mothers with a child less than 12 months of age and mothers with more than one child. No other key differences were observed between the age ranges and number of children. We have added to the sentence (page 30, lines 698-700 on the marked-up copy) to improve clarity. 

It is still possible to quantify aspects of a qualitative study:

• We note this suggestion and have added further percentages in the results section (page 21-21, line 481 and page 22, line 509 on the marked-up copy).

I would love to see more pointed recommendations for those trying to combat AMR. There were important findings such as the use of the internet, the sharing of medications, etc. It would be helpful to do a f/u study of providers to understand their perspective and experience prescribing antibiotics in this setting.

• We agree with the reviewer that more pointed recommendations about addressing AMR are important as well as a future follow-up research with prescribers. Thus, further details have been added to the discussion section to include these suggestions (page 31, lines 726-739 on the marked-up copy).

Reviewer 2:

The abstract should end with the implications of the results. Apart from that, the abstract is well-structured and focused:

• We have added a sentence to the end of the abstract to include the implications of the findings, as suggested by the reviewer. Page 3, lines 57-59, on the marked-up copy.

The introduction is too long and difficult to read. It should be structured into 3 paragraphs. The first paragraph should highlight the background of the research, from a global to a more local perspective. The second paragraph should demonstrate the current state of the research topic, putting it into context with other research results around the world. The last paragraph should highlight what is yet to be known about the research topic and what the main objective of the study is:

Academic Editor: It is good to frame the introduction section to make it better as suggested but not limited to three paragraphs: 

• We agree with the comments in order to improve the introduction and have made amendments. The introduction has been shortened and sections re-written, but has not been limited to three paragraphs. Please see changes from page 3, lines 62-113 (on the marked-up copy). 

It is explained that due to COVID-19, the focus groups were conducted via zoom. I believe this should be discussed as a strength or weakness of the design, rather than using it as a justification for conducting the focus group interviews:

• We agree with the reviewer’s comments and the online design is now discussed further in the limitations section, page 32, lines 748-750 (on the marked-up copy). 

In the data analysis section, was thematic analysis done with any software? In addition, it is explained that 2 researchers conducted the analysis, but did they do it independently? How did they resolve interpretive discrepancies?

In the same section, it should be informed how the focus groups were concluded. Were they ended when information saturation was reached?

• Qualitative analysis software was not used to organise the data for analysis and Microsoft Excel was used to input data codes. All details regarding the computer software used, the procedure for performing data analysis, who conducted the analysis and how issues were resolved is outlined in the Supporting Information – Data Analysis Protocol. This has been updated to improve clarity in response to the queries raised by the reviewer. We agree with the reviewer’s comments that it should be specified how focus groups were concluded and we have updated this in the Data Collection section to specify that sessions finished when saturation was reached, page 10, line 236 (on the marked-up copy). 

Conclusions: It should be shortened. It is too long. It should be summarized in one conclusion. Additionally, it would reinforce that the main implications of the study should be discussed in this section.

• The conclusion has been reduced based on the reviewer’s feedback, page 33, lines 773-775 and 785-788 (on the marked-up copy).

Thank you for considering the suitability of this manuscript further. 

Yours sincerely

Professor Mitchell Byrne

Faculty of Health

Charles Darwin University

Email: mitchell.byrne@cdu.edu.au

Tel: 0417 859 559

---

## [Decision Letter · Decision Letter 1]

28 Jun 2023

What influences parental decisions about antibiotic use with their children: A qualitative study in rural Australia

PONE-D-23-05601R1

Dear Dr. Mitchell K Byrne,

We’re pleased to inform you that your manuscript has been judged scientifically suitable for publication and will be formally accepted for publication once it meets all outstanding technical requirements.

Kind regards,

Kshitij Karki, MPH, MA

Academic Editor

PLOS ONE

Additional Editor Comments (optional):

Reviewers' comments:

Reviewer's Responses to Questions

**Comments to the Author**

1. If the authors have adequately addressed your comments raised in a previous round of review and you feel that this manuscript is now acceptable for publication, you may indicate that here to bypass the “Comments to the Author” section, enter your conflict of interest statement in the “Confidential to Editor” section, and submit your "Accept" recommendation.

Reviewer #1: All comments have been addressed

2. Is the manuscript technically sound, and do the data support the conclusions?

Reviewer #1: Yes

3. Has the statistical analysis been performed appropriately and rigorously? 

Reviewer #1: Yes

4. Have the authors made all data underlying the findings in their manuscript fully available?

Reviewer #1: Yes

5. Is the manuscript presented in an intelligible fashion and written in standard English?

Reviewer #1: Yes

6. Review Comments to the Author

Reviewer #1: (No Response)

7. PLOS authors have the option to publish the peer review history of their article (what does this mean?). If published, this will include your full peer review and any attached files.

Reviewer #1: No

---

## [Editor Report · Acceptance letter]

11 Jul 2023

PONE-D-23-05601R1 

What influences parental decisions about antibiotic use with their children: A qualitative study in rural Australia 

Dear Dr. Byrne:

I'm pleased to inform you that your manuscript has been deemed suitable for publication in PLOS ONE. Congratulations! Your manuscript is now with our production department. 

Kind regards, 

on behalf of

Dr. Kshitij Karki 

Academic Editor

PLOS ONE